# Variations of organic functional chemistry in carbonaceous matter from the asteroid 162173 Ryugu

Primordial carbon delivered to the early earth by asteroids and meteorites provided a diverse source of extraterrestrial organics from pre-existing simple organic compounds, complex solar-irradiated macromolecules, and macromolecules from extended hydrothermal processing. Surface regolith collected by the Hayabusa2 spacecraft from the carbon-rich asteroid 162173 Ryugu present a unique opportunity to untangle the sources and processing history of carbonaceous matter. Here we show carbonaceous grains in Ryugu can be classified into three main populations defined by spectral shape: Highly aromatic (HA), Alkyl-Aromatic (AA), and IOM-like (IL). These carbon populations may be related to primordial chemistry, since C and N isotopic compositions vary between the three groups. Diffuse carbon is occasionally dominated by molecular carbonate preferentially associated with coarse-grained phyllosilicate minerals. Compared to related carbonaceous meteorites, the greater diversity of organic functional chemistry in Ryugu indicate the pristine condition of these asteroid samples.

The low-albedo asteroid 162173 Ryugu is one of many C-type asteroids[1–3] that are thought to be parent bodies of carbonaceous chondrite meteorites that fall to Earth. Ryugu's current form of a top-shaped, rubble-pile asteroid was created by a previous collisional disruption of a larger body[3,4]. However, orbital near-infrared spectroscopy detected a weak but ubiquitous 2.72 μm absorption feature indicative of hydrated minerals[5], consistent with a pre-collisional history of hydrothermal aqueous processing. Preliminary examination of Hayabusa2 return samples by the JAXA curation team[6,7] and the Hayabusa2 Initial Analysis Teams[8,9] indicated petrographic, composition, and isotopic similarities to Ivuna-type (CI) carbonaceous chondrites. This group of rare chondritic meteorites are notable in that their bulk compositions closely match that of the solar photosphere, establishing a valuable reference material for the average isotopic and elemental composition of the early solar nebula. However, all CI chondrite samples show evidence of extensive aqueous alteration on their parent asteroid(s)[10,11], and although the presence of extraterrestrial organic molecules has been demonstrated in these meteorites[12–14], the question of how much of this alteration may be due to terrestrial contamination and weathering has not been resolved[15–17].

The Hayabusa2 spacecraft collected surface regolith particles from Ryugu in two separate touchdown events, which were stored in collection chambers A and C of the spacecraft's sample catcher[18]. These samples were protected within the Hayabusa2 re-entry capsule from atmospheric re-entry heating upon Earth arrival and have been stored under vacuum or an ultra-purified nitrogen atmosphere while at the JAXA curation facility. In this study, we investigate the functional group chemistry of discrete carbonaceous grains and diffuse carbonaceous matter within these particles. We analyzed three types of sample preparations: insoluble organic matter (IOM) residues extracted from particles by an HF/HCl dissolution procedure, thin sections of particle fragments prepared by ultramicrotomy, and thin sections extracted by focused ion beam (FIB) (Methods). A corresponding set of samples were also prepared from IOM and fragments of the CI meteorite Orgueil. While IOM residues provide a representative aggregated sampling of carbonaceous matter from each collection site on Ryugu, they are missing both soluble organic matter and valuable information regarding the local petrologic context with surrounding mineral grains. This missing context is provided by the ultramicrotome and FIB sections, although these preparations are less representative

e-mail: bradley.t.degregorio.civ@us.navy.mil

of Ryugu regolith. Comparisons between microtome and FIB sections are necessary to identify soluble organic matter and potential adverse effects of electron beam exposure to Ryugu carbonaceous phases from FIB.

The vast majority of the organic matter in carbonaceous chondrites is composed of a complex macromolecule made of interconnected polyaromatic moieties and aliphatic side chains[19–21]. X-ray absorption near-edge structure (XANES) spectroscopy is useful for characterizing such material, primarily because double- and triple-bonded carbon (e.g., C=C, C=O, C≡N) has large photoabsorption oscillator strengths for electronic transitions from core shell (1$s$) states to hybridized π* anti-bonding states[22]. Bulk XANES studies of IOM from many carbonaceous chondrites share common spectral features[23], namely three peaks arising from aromatic C=C ring structures in polyaromatic domains, ketone (C=O) functional groups attached to aliphatic side chains, and carboxyl (COOH) functional groups attached to either aromatic or aliphatic carbons. Aliphatic carbons can also be detected by this method, although the oscillator strength for the corresponding 3$p$/σ* anti-bonding states is much weaker than their π* counterparts[24]. Thus, the presence of aliphatic features in carbon XANES spectra implies a high abundance of aliphatic carbon relative to π-bonded carbon groups. Small variations in the relative intensities of the main π* peaks observed between different meteorite group are consistent with the overall degree of alteration and heating of their respective asteroid parent bodies, with more aqueously altered chondrites containing decreasing ketone and carboxyl peaks, and heated chondrites containing increased aromatic peaks[23,25]. At the μm-to-sub-μm scale, the organic functional chemistry of IOM can be heterogeneous[26]. For example, some larger, discrete IOM grains occasionally show highly aromatic functional chemistries consistent with a higher concentration of slightly larger polyaromatic domains and shorter aliphatic side chains. Using analogous, higher spatial resolution methods in transmission electron microscopy (TEM), these functional group heterogeneities at scales down to a few nm have been observed within single carbonaceous grains[27]. Diffuse organic matter observed in situ in chondrite samples with XANES and TEM can show even more heterogeneous functional group compositions than those found in IOM samples[25,28,29].

Here we apply these methods to assess the diversity of carbonaceous matter in Ryugu IOM samples A0106 and C0107 and intact particles from Ryugu aggregate samples A0108 and C0109. Correlated XANES, STEM, and nanoscale secondary ion mass spectrometry (NanoSIMS) of these samples reveals the morphology and chemical characteristics of different populations of carbonaceous material, including discrete carbon grains and diffuse carbonaceous matter (Supplementary Fig. 1). Shape analysis of XANES spectra using Gaussian fitting and a hierarchical clustering algorithm found three distinct functional chemistry populations. In addition, a unique molecular carbonate phase is present in FIB-prepared sections intermixed with coarse-grained phyllosilicates. Carbonaceous matter in both more abundant and more diverse than that in the CI chondrite Orgueil, indicating that asteroidal material is more pristine than its meteoritic counterparts on earth and holds promise to reveal new information about components and processes from the early solar system.

## Results

IOM samples prepared from both Ryugu collection chambers show XANES characteristics similar to IOM from aqueously altered chondrite meteorites. "Bulk" spectra are similar to those of CI and CM (Mighei type) chondrites, both of which are among the most aqueously altered meteorite groups. However, a significant number of discrete, compact, grains have XANES spectra that deviate from bulk spectra (Fig. 1). We used spectral decomposition with a series of Gaussian peaks and a hierarchical clustering algorithm to perform a shape analysis and identify similarities within the spectral dataset (Methods;

Supplementary Fig. 2). Shape analysis of XANES spectra from 23 discrete grains (Fig. 1D) indicate that twelve (about 50%) have IOM-like (IL) spectra, containing the three typical peaks, with most spectra showing slightly sharper ketone and carboxyl peaks (Fig. 1E). This slight difference in C=O features between bulk IOM and individual discrete grains was seen previously for nanoglobules within chondritic IOM residues[26]. Another seven (~30%) of these IOM grains show highly-aromatic (HA) spectra with a characteristic broadening of the aromatic C=C peak and minimal ketone and carboxyl peaks. Heirarchical clustering identified a group of four (~20%) Ryugu IOM grains distinct from the typical IL and HA spectral shapes. It consists of an intense but narrow aromatic peak and a subtle bump in the 287–288 eV range (Fig. 1E and Supplementary Fig. 3). The narrow aromatic peak centered at 285.0 eV indicates a more limited distribution of aromatic functionality, most likely an increase in the amount of single aromatic rings and small polyaromatic domains, as opposed to the larger polyaromatic moieties and broader aromatic peak observed in the highly aromatic grains. The greater 287–288 eV absorption, however, could be caused by a greater abundance of aromatic ring-bound carboxyl functional groups, amide groups, or aliphatic moieties[22,24]. Our finding that the XANES spectra from discrete grains cluster into three main spectral shapes matches the preliminary results from the Hayabusa2 Initial Analysis, where the AA spectral shape is identified in that study as aromatic[8]. Almost all N-XANES spectra of carbonaceous grains from each of the three spectral shapes are flat or contain a featureless bump (Supplementary Fig. 4), indicating low N abundance, in keeping with previous observations of CI and CM IOM (where N/C is typically ~3 at.%[30]). A non-detection of nitrogen with XANES has been shown to have a typical upper bound of N/C < 2 at.% [31]. Two out of three total N-XANES spectra from HA grains contain clear N absorption edges, suggesting preservation of some N in polyaromatic heterocycles, as observed in N-bearing nanoglobules in other carbonaceous chondrites[26]. However, the lack of an observed increase in N absorption in this third spectral type over HA or IL spectra (Supplementary Fig. 4) suggests contribution from amide functionality is unlikely. O-XANES spectra of Ryugu IOM, IL grains, and this third spectral type show a similar intensity ratio between the 531 eV π* absorption from carbonyl C=O functional groups and the σ* absorption at 540 eV from C-O single bonds[32] (Supplementary Fig. 4), suggesting relatively similar abundances of O composition. In contrast, HA grains show a smaller π*:σ* peak ratio due to a smaller abundance of O-bearing functional groups. Thus, we tentatively describe this new alkyl-aromatic (AA) material as consisting of an increased amount of smaller polyaromatic domains, predominantly single or double benzene rings, decorated predominantly by carboxyl functional groups. This molecular interpretation is similar to that of Ryugu IOM and IL grains but with a limited range of small polyaromatic domains and longer aliphatic chains connecting them. This population of alkyl-rich macromolecular material could be related to the abundant alkylated polycyclic aromatic hydrocarbons (such as alkylbenzene) detected in Ryugu samples[33].

This diversity in organic functional chemistry is also observed in microtome and FIB sections from Ryugu particles (Fig. 2). Carbonaceous material is found amidst fine-grained matrix material in x-ray photoabsorption maps. Discrete, compact grains are observed, including some with nanoglobule morphologies. Shape analysis of C-XANES spectra from these grains (Supplementary Fig. 5) shows that they fall into the three previously-observed spectral shape categories described above for Ryugu IOM. Heirarchical cluster analysis of 38 discrete grains in these sections determined a HA:AA:IL spectral shape ratio of ~20:25:55, similar to that of the IOM samples. Plotting the spectral contributions from the major absorption features in these spectra show a clear distinction between HA and IL spectral shapes, as well as the related cluster of AA grains (Fig. 2E). The overlap between the AA and IL fields in this plot illuminate how similar these two

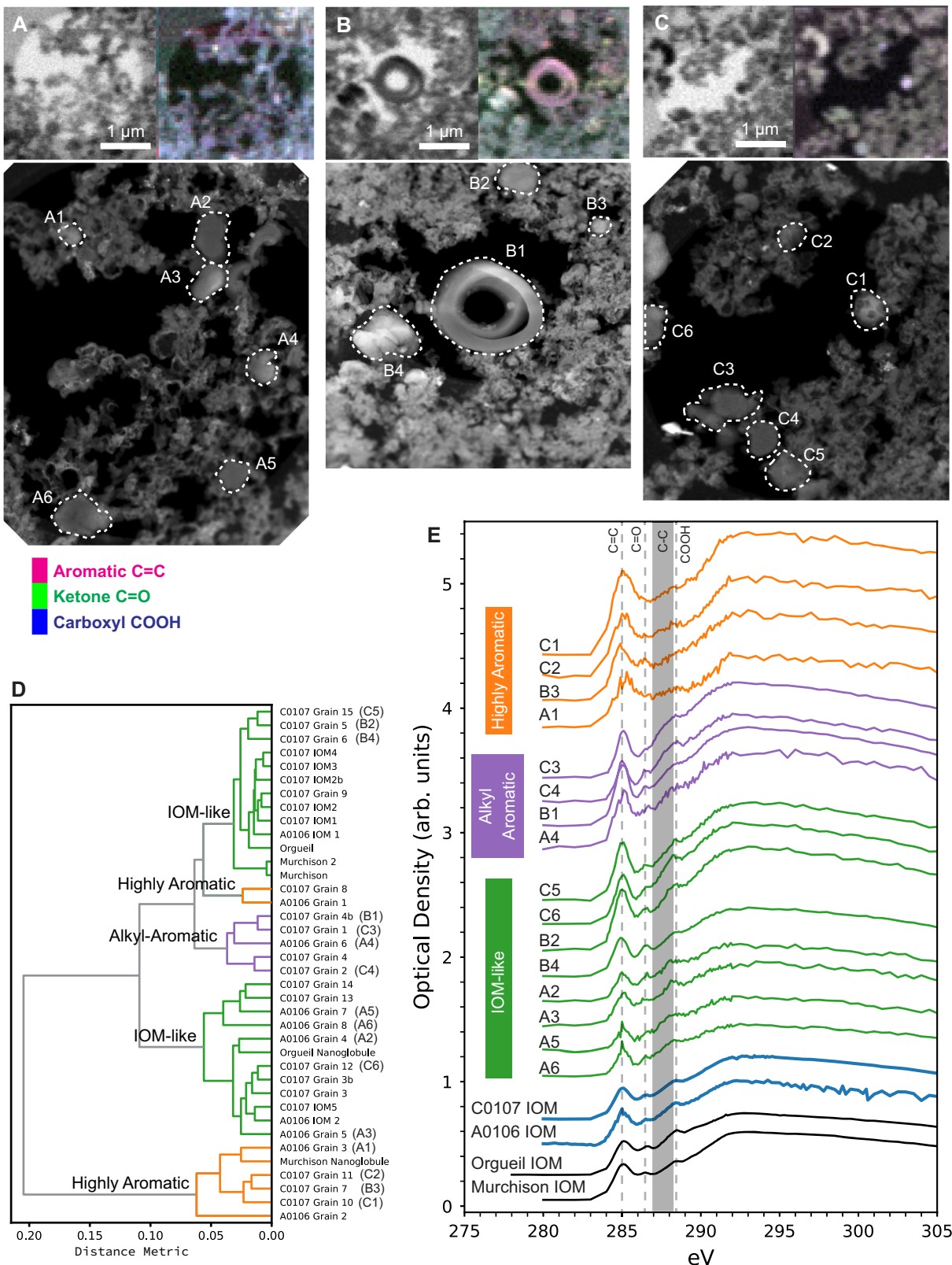

**Fig. 1 | Scanning transmission x-ray microscoy (STXM) and scanning transmission electron microscopy (STEM) imaging and x-ray absorption near-edge structure (XANES) spectroscopy of Ryugu insoluble organic matter (IOM).** Each set of images in **A**–**C** are a correlated STXM image at 290 eV, a false-color map of organic functional chemistry (magenta = aromatic C=C at 285 eV; green = ketone C=O at 286.7 eV; blue = carboxyl COOH at 288.5 eV), and a high-angle annular dark field (HAADF) STEM image. **A** IOM from Chamber A sample A0106. **B**, **C** IOM from Chamber C sample C0107. **D** Hierarchical clustering dendrogram of fitted XANES

spectra from Ryugu IOM samples A0106 and C0107. The length of horizontal lines indicates the distance between spectral locations in a 21-dimensional space determined by Gaussian peak fitting. Samples labeled with IOM denote average spectra of surrounding IOM in the vicinity of the discrete grains (excluding other discrete grains). **E** XANES spectra from individual discrete grains and nanoglobules in **A**–**C**, compared to bulk, fluffy-textured IOM from A0106 and C0107, as well as IOM from the Orgueil (CI) and Murchison (CM) carbonaceous chondrites.

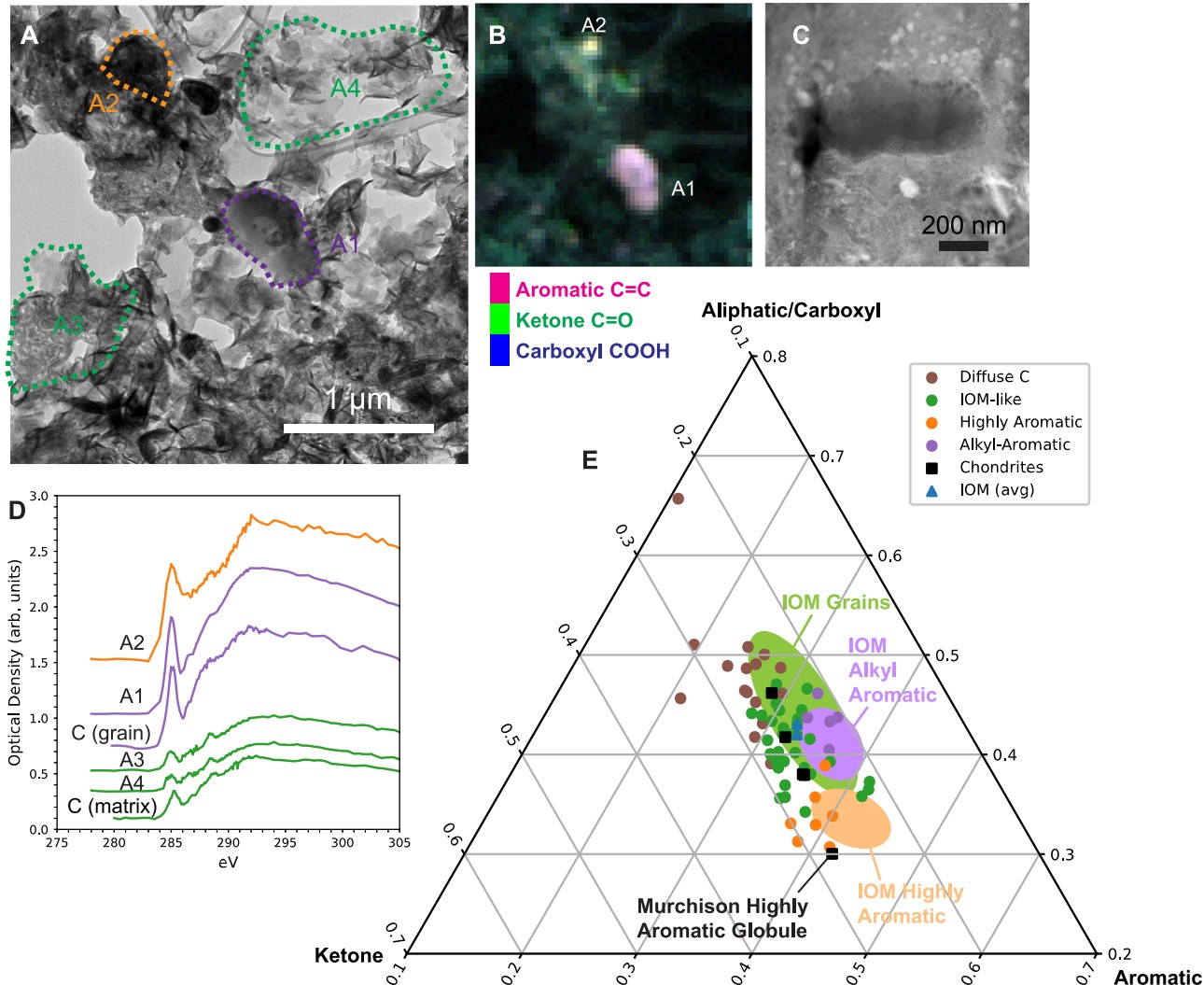

**Fig. 2 | Transmission electron microscopy (TEM) and Scanning transmission x-ray microscopy (STXM) imaging and x-ray absorption near-edge structure (XANES) spectroscopy of carbonaceous grains in Ryugu fine-grained matrix. A** TEM image of sample C0109-11[8] containing two carbonaceous grains (A1 and A2). **B** STXM false-color map of the same region showing the distinct organic functional chemistry of the two grains (magenta = aromatic C=C at 285 eV; green = ketone C=O at 286.7 eV; blue = carboxyl COOH at 288.5 eV). The surrounding phyllosilicate-rich matrix also contains diffuse C, visible as green and green-blue regions in this map. **C** High-angle annular dark field (HAADF) scanning TEM image of an alkyl-aromatic globular grain in a focused ion beam (FIB) section extracted from Ryugu particle A0108-3 (see Fig. 3). **D** XANES spectra of carbon grains and matrix shown in **A–C. E** Distribution of characteristic absorption peak area (plotted as a proportion of the total near-edge structure; see Methods) for carbonaceous grains and diffuse C in ultramicrotome and FIB sections. Solid color fields show the distributions for carbon grains identified in Ryugu insoluble organic matter (IOM) samples.

spectral shapes are to the human observer, and other non-traditional representations of C-XANES spectra may be necessary to distinguish them (e.g., Supplementary Fig. 7). Considering that these in situ sections are a less representative sampling of Ryugu than the IOM residues, these data suggest little difference in the diversity of functional chemistry between discrete carbonaceous grains in intact Ryugu particles and Ryugu IOM. In contrast, C-XANES measurements from 18 discrete carbonaceous grains in ultramicrotome sections from Orgueil matrix particles did not reveal any HA or AA spectral shapes, but only produced spectra consistent with IL functional chemistry (Supplementary Fig. 6), consistent with a previous XANES study of nanoglobules in Orgueil IOM[26], although the low counting statistics ($N = 4$) in that study cannot rule out the presence of grains with HA or AA spectral shapes in Orgueil.

Correlated STXM, STEM, and NanoSIMS-based N and C isotopic measurements of carbonaceous grains in a FIB liftout lamella extracted from Chamber A particle A0108-3 may suggest a connection between their functional chemistry and isotopic composition.

Previous studies of Ryugu samples have indicated a large spread of [15]N, [13]C, and D values, but not correlated with functional chemistry of the organic matter. Here we report C-XANES data of 15 relatively large carbonaceous grains for which N and C isotopic compositions were previously measured (Fig. 3)[8]. Most of the measured carbonaceous grains in this section (small gray dots in Fig. 3E), identified by concentrations of carbonaceous matter in NanoSIMS image frames, have C and N isotopic compositions indistinguishable from the "bulk" isotopic composition of Ryugu organic matter[8,34], within errors. However, most of the large carbonaceous grains deviate by >2σ in at least one of these isotopic systems (Supplementary Table 1). Both the highest [15]N and [13]C enrichments are observed in grains with an IL functional group chemistry, while the two measured grains with HA functional chemistry contain depletions in both [15]N and [13]C. Similar simultaneous large isotopic depletions in both [15]N and [13]C were previously observed in Ryugu carbonaceous grains[8,35]. In carbonaceous meteorite samples, correlated [15]N and [13]C depletions have been observed in carbonaceous grains within many primitive chondrites,

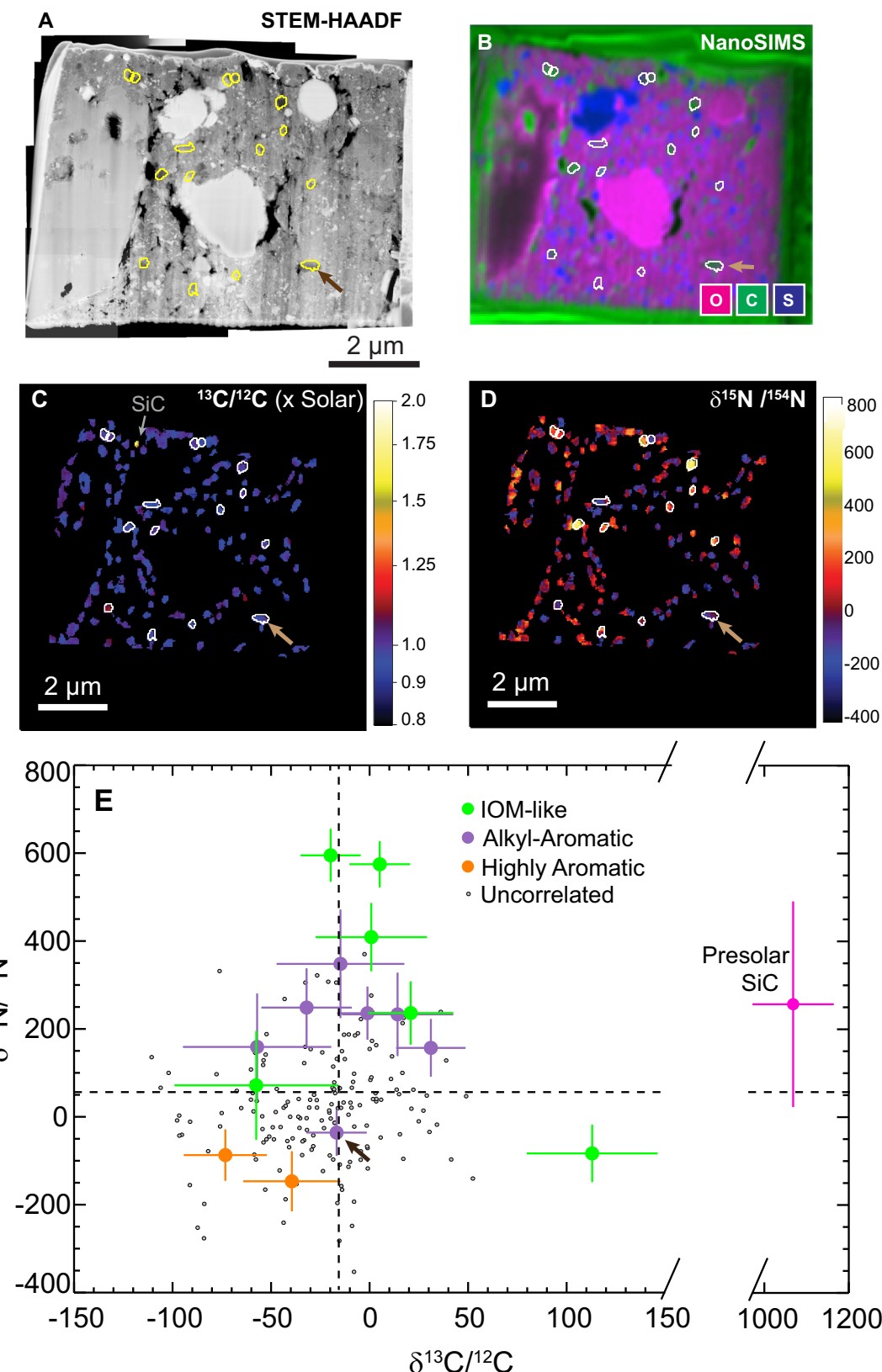

**Fig. 3 | Isotopic composition of Ryugu carbonaceous grains. A** High-angle annular dark field scanning transmission electron microscopy (HAADF-STEM) mosaic of a focused ion beam (FIB) lamella extracted from Chamber A particle A0108-3. **B** Nanoscale secondary ion mass spectrometry (NanoSIMS) elemental map of O, C, and S (in magenta, green, and blue, respectively). **C** Isotope ratio image of $^{13}C/^{12}C$ normalized to the solar value, also showing the location of a presolar SiC grain. **D** Isotope ratio image of $\delta^{15}N/^{14}N$. Locations of carbonaceous grains in the section for which both x-ray absorption near edge structure (XANES) spectra and isotopic data were acquired are outlined in **A**–**D**. **E** C and N isotope compositions of carbonaceous grains. Grains with corresponding XANES spectra are color-coded by their shape classification. Dashed lines indicate "bulk" values for Ryugu[34]. Carbon grains that are not correlated with scanning transmission x-ray microscopy (STXM) data (gray points) have error bars (2σ) similar in size to the colored points, but have been removed for clarity. The arrows in (A-E) indicate the location and isotopic composition of the carbon grain shown in Fig. 2C.

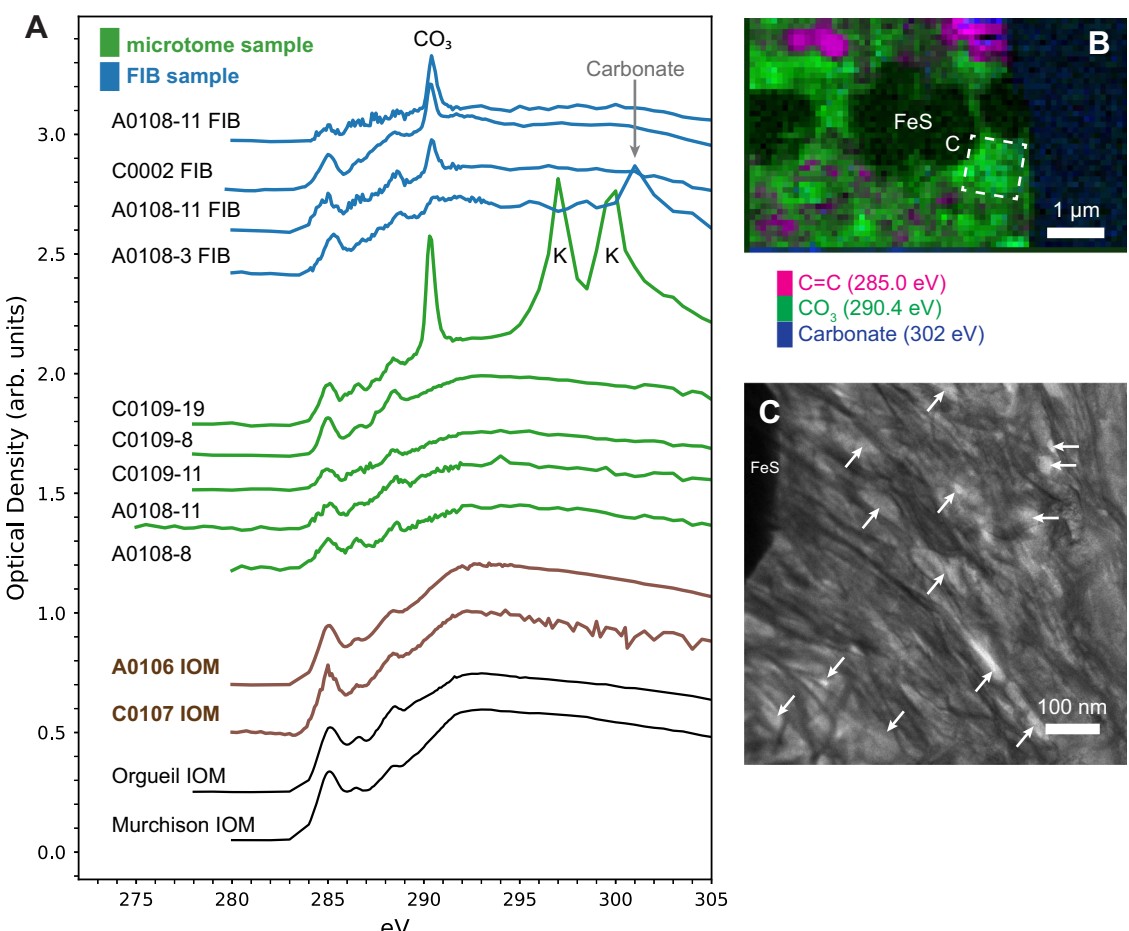

**Fig. 4 | Diffuse carbonaceous matter in Ryugu. A** X-ray absorption near-edge structure (XANES) spectra of Ryugu diffuse organic matter in ultramicrotome and focused ion beam (FIB) lamellae, compared with bulk Ryugu, Orgueil, and Murchison insoluble organic matter (IOM). **B** Scanning transmission x-ray microscopy (STXM) spectral map of a FIB lamella from Chamber A grain A0108-11 showing concentrations of diffuse organic matter with a molecular carbonate signature (magenta = carbon at 285 eV; green = carbonate $CO_3$ at 290.4 eV; blue = mineral carbonate at 302 eV). Mineral carbonates would be visible as a teal color in this map. **C** Bright-field scanning transmission electron microscopy (STEM) image of intermixed phyllosilicate sheets and brighter, interstitial organic matter from the region shown in **B**. Arrows denote locations of accumulations of $CO_3$-rich organic matter.

including those that have experienced little aqueous alteration[36,37] and other chondrites where aqueous alteration was more extensive[38]. While these data are very limited, they suggest that carbonaceous grains with different functional chemistry may compose distinct reservoirs of isotopic compositions, most likely reflecting distinct origins.

Diffuse organic matter is abundant in the fine-grained matrix of Ryugu[8], visible in carbon STXM maps of microtome and FIB sections (e.g., Fig. 2B). The functional group chemistry of this diffuse organic matter varies from sample to sample but is usually consistent from region to region within a given section, suggesting a pervasive influence of parent body aqueous fluids. In previous XANES studies of CI chondrite meteorites, diffuse carbon is characterized by a dominant carboxyl functional group absorption, an absent or minor ketone peak, a minor amount of aromatic photoabsorption, and the occasional presence of a new peak at 290.4 eV assigned to $CO_3$ functional groups[25]. Ryugu diffuse carbon is different in that most instances show a more intense aromatic peak and a clear ketone peak (Fig. 4). In all of the analyzed FIB sections in this study and one of the ultramicrotome sections, a 290.4 eV photoabsorption is also observed. XANES spectral mapping of this 290.4 eV peak in FIB and microtome sections reveals relatively large regions of diffuse organic matter (Fig. 4B), which is not similarly observed in CI or CM chondrites. For the FIB section from Ryugu sample A0108-3, this peak is accompanied by characteristic,

strong, extended fine structure from carbonate minerals. This extended fine structure is present in a calcite grain in the section and within organic matter in the immediate surrounding mineral matrix, presumably taking the form of nanoscale calcite grains. Therefore, the 290.4 eV peak can confidently be assigned to $CO_3$ functional groups in carbonate. In the other samples, however, no carbonate minerals were observed and no extended fine structure is present, suggesting this photoabsorption could be due to organic, molecular $CO_3$. An abundant molecular carbonate phase was also previously reported in Ryugu matrix[8,39]. O-XANES spectra of this material lack a C=O peak, but this could be due to an overwhelming effect from the encompassing O-rich phyllosilicate groundmass.

Diffuse molecular $CO_3$ in the Ryugu samples appears to be associated with phyllosilicate minerals instead of carbonates[39]. Spectral maps of the $CO_3$ photoabsorption at 290.4 eV show a correlation between absorption and the distribution of phyllosilicates, with the greatest concentrations found where coarse-grained phyllosilicates were present, confirmed by subsequent high resolution STEM imaging (Fig. 4B, C). Other XANES studies of Ryugu organic matter also noted a distinct functional group composition in association with asteroidal phyllosilicates[7,8], and their published spectra also contain a small $CO_3$ peak at 290.4 eV. While previous XANES data from carbonaceous chondrite meteorites has established a spatial relationship between diffuse carbon and phyllosilicates arising from hydrothermal

alteration of primary accreted amorphous silicate grains[40,41], one TEM-based study of phyllosilicate minerals from the CI chondrites Orgueil and Ivuna, as well as the ungrouped carbonaceous chondrite Tagish Lake, also detected a strong $CO_3$ energy-loss peak from isolated clay mineral grains[42]. This $CO_3$ peak was not observed in carbonaceous aggregates with less developed phyllosilicate grains, leading the authors to conclude that coarse-grained clay minerals in CI chondrites may contain significant amounts of molecular $CO_3$ bound to clay sheet surfaces and/or interlayer sites[42]. Our observations of the distribution of the XANES $CO_3$ signal across the in situ Ryugu samples is consistent with this latter, clay-bound interpretation, and visual TEM confirmation of organic matter in between phyllosilicate sheets (Fig. 4C) indicates the presence of an abundant clay-associated polycarbonate material in Ryugu. While this spectral feature is prevalent in C-XANES data acquired from FIB sections, it is observed less frequently in ultramicrotome sections. The molecular $CO_3$ signal appears preferentially spatially associated with clay mineral surfaces, but not prevalent in organic matter a few nm from the clays. The FIB samples have a relatively uniform thickness (-100 nm), providing much more clay volume, and hence more bound $CO_3$ x-ray photoabsorption. In contrast, clay minerals in ultramicrotome sections have variable thickness, ranging from relatively isolated flakes only a few nm thick to phyllosilicate stacks up to 70 nm (i.e., the slice thickness). In addition, no molecular $CO_3$ functionality was observed in Ryugu IOM samples, in which the clays and soluble organics have been removed.

## Discussion

The combined observations of discrete carbonaceous grains and diffuse organic matter in Ryugu particles is broadly consistent with the functional chemistry and distribution of carbonaceous matter in CI and CM chondrites[7,8]. That is, Ryugu organic matter is characteristic of asteroidal organic matter that has experienced a high level of aqueous alteration at relatively low temperatures (<150 °C). However, Ryugu organic matter differs from CI and CM organic matter in three key aspects. First, Ryugu organic matter overall appears to be more aromatic than that of CI and CM chondrites. Ryugu samples contain a higher proportion of carbonaceous grains with a HA or AA spectral shape, and C-XANES spectra from Ryugu diffuse carbon generally contain greater aromatic photoabsorption than their counterparts in CI and CM samples[25,26,29]. Second, previous STXM and TEM studies of carbonaceous matter from the Orgueil, Murchison, and Paris meteorites have reported organic grains with functional chemistries that fall into the IL and HA spectral shapes, but not AA spectral shapes[25,26,29]. This may reflect the relatively small number of organic grains analyzed, but alternatively, this AA material may represent organic evolution unique to aqueous alteration and mineral composition of the Ryugu parent body. Application of automated spectral shape analysis to future studies of organic matter in CI chondrites may yet discover examples of AA grains in these Ryugu analog samples. Thirdly, as discussed above, compared with existing data from CI chondrites, we find $CO_3$ functional groups are much more prevalent in Ryugu diffuse carbon. In a few of the Ryugu samples, $CO_3$ functionality dominates the average functional chemistry of diffuse carbon over regions several μm in size. Our understanding of Ryugu organic matter is still in its infancy, and more analyses of Ryugu samples are needed to support whether or not these conclusions hold generally for the Hayabusa2 collection. In addition, the number of in situ studies of CI and CM chondrites that focus on diffuse carbon is small, and could present a sampling bias against a similar abundance of $CO_3$-rich diffuse carbon in those samples. Other studies have reported that Ryugu organic matter may contain more abundant aliphatic functionality[7] than CI and CM meteorites, in opposition to our observations. Analysis of released gases from the sample collection chambers A and C indicate that

Ryugu may be comparatively deficient in N content[43], although our data indicate that the N composition of Ryugu organic matter is similar to that of CI and CM meteorites.

These general differences in functional chemistry of macromolecular carbon between Ryugu and other sibling meteorites in the CI chondrite group can be explained by differences in the extent of low temperature aqueous alteration on C-type asteroid parent bodies. One point of comparison is with a comprehensive study of IOM extracted from several petrologic subunits of the ungrouped carbonaceous chondrite Tagish Lake[44], predicted to be sourced from a D-type asteroid[45]. A clear alteration trend exists in the elemental composition, molecular and functional group diversity, and isotopic composition of organic matter across the five petrologic sub-units. The most pristine sub-unit (Tagish Lake sample 5b) contains the lowest concentration of polyaromatic moieties and highest D/H isotopic composition. The remaining subunits show evidence of progressive aqueous alteration, correlated with a gradual change in the properties mentioned above, with the most altered subunits (Tagish Lake samples 11i and 11v) containing the highest concentration of polyaromatic moieties and the lowest D/H ratio[44]. Interestingly, nitrogen isotopes appear to be unaffected, with similar $\delta^{15}N$ compositions observed across the Tagish Lake subunits. These trends are further supported by TEM and XANES studies of CR chondrites and the CM chondrites Murchison and Paris, which observed a similar increase in polyaromatic bonding in the more altered Murchison samples[28,29]. While it is unknown if the actual mechanisms and chemical reaction networks for aqueous alteration of organic matter is the same across these three meteorite groups (i.e., CIs, CMs, and Tagish Lake, presumably equivalent to C-type, B-type, and D-type asteroids), the general trend of increasing aromaticity with progressive aqueous processing in carbonaceous chondrites has been established[46].

Differences between Ryugu organic matter and that found in related CI chondrite meteorites like Orgueil may not just be due to variable levels of aqueous processing. The pristine collection of material from the Ryugu surface by Hayabusa2 successfully avoids additional shock, heating, and alteration due to impact ejection from the asteroid surface, increased exposure to space radiation environments during travel through the Solar System, atmospheric entry heating, terrestrial impact, and resident exposure to the wet terrestrial environment. For example, heating simulations and experiments on amino acids and nucleobases commonly found in carbonaceous chondrites indicate a survival rate of 10% or less during impact ejection events and atmospheric entry heating[47,48]. It has also been suggested that exposure to terrestrial atmosphere can oxidize and degrade organic matter preserved in interlayer sites within phyllosilicate minerals[49], but this effect may also destroy other labile organics within the Ryugu fine-grained matrix. While the more refractory carbonaceous matter can withstand these exposures better than soluble organic molecules, alteration of the macromolecular structure, composition, and functional group diversity of this material is highly likely. The ubiquitous presence of diffuse organic matter in Ryugu fine-grained matrix, relative to that found in meteorite samples, supports this hypothesis. In this regard, the Hayabusa2 collection is the most pristine representation of organic matter on C-type asteroids, and preservation of organic components for future investigations will require protection from heating and oxidation. Analyses of organic matter from asteroid sample return missions, such as OSIRIS-REx, may also expect increased organic functional group diversity in their samples. While the CI chondrites still provide an appropriate sample set for investigating exogenous delivery of organics to planetary bodies, samples from the Hayabusa2 and OSIRIS-REx collections may soon become the principal resource for the study of the formation and modification of pre-accretionary chondritic organic matter in the early Solar nebula.

# Methods

## IOM extraction

Residues of insoluble organic matter were prepared from Ryugu grains by acid demineralization after solvent-extraction of soluble organic matter. The samples were treated in a stirred 6 N HCl solution with high-purity water in Teflon microtubes for ~48 h, followed by a 9 N HF/1 N HCl treatment for >48 h. This process was repeated three times, interspersed with 1 N HCl rinses, high-purity water rinses, and a final methanol rinse. The final acid-insoluble, carbonaceous residues were transferred to glass vials and dried at 55 °C on a hotplate. The glass vials were pre-heated to 500 °C for 4.5 h in air prior to use. IOM samples of Murchison and Orgueil used in this study were sourced from Alexander et al.[50].

## Ultramicrotomy and focused ion beam preparation

Intact grains from Ryugu, meteorite analogs, and IOM powders were transferred with freshly-made glass needles and placed in molten droplets of sulfur kept at 114 °C on a heated glass plate. Needle micromanipulation and monitoring/recording of this operation was done using an EXpressLO ex situ liftout system. Once the heat source was removed, the sulfur crystallized around the grains, and the entire droplet was attached to an 8 mm epoxy block with a thin layer of cyanoacrylate adhesive. Ultramicrotomy was performed using a diamond knife in a Leica EM UC6 microtome. Sections were created at 70 nm thicknesses and placed on lacey carbon (for intact grains) or holey carbon (for IOM) support films on 200 mesh copper TEM grids for both STXM and TEM analysis.

Focused ion beam (FIB) liftout sections in this study were prepared at the U.S. Naval Research Laboratory with a Thermo Fisher Helios G3 DualBeam FIB and at the University of Tokyo with a Hitachi FB-2100 FIB. At NRL, Ryugu grains were transferred with freshly-made glass needles and deposited on double-sided copper sticky tape on a standard 12.5 mm pin mount. At UT, Ryugu grains were pressed into a 3 mm gold disk with a diamond-stainless steel jig press. The gold disks were pre-cleaned with methanol/dichloromethane and Milli-Q water and then heated at 500 °C for 3 h. Sample mounts were coated with ~25 nm of amorphous carbon or tungsten to minimize charging. Both FIB-SEM systems used 30 keV $Ga^+$ ions to sputter away material, using typical FIB liftout protocols. with the exception that no electron imaging was performed on the lamellae after their thickness was less than 1.5 μm to avoid possible radiolysis damage to organic matter by the 5 keV electron beam[51]. Ion-assisted carbon deposition was used for the protective cover layer, which is easily recognizable with STXM and TEM. Lamellae were attached to copper half grids with ion-assisted platinum deposition before thinning to a final thickness of <100 nm. Once an adequate sample thickness was achieved, the lamellae were given a final cleaning step of 8–10 keV $Ga^+$ ions on each side. Sections prepared at UT were further polished with a low energy Ar ion milling system (Fischione NanoMill 1040) at energies of 2 kV, 900 V, and 500 V.

## Scanning transmission X-ray microscopy

STXM measurements were acquired at both beamline 19A at Photon Factory (PF), High Energy Accelerator Research Organization (KEK), Japan and beamline 5.3.2.2 at the Advanced Light Source, Berkeley, CA, USA. The synchrotron radiation X-ray source at compact STXM beamline 19A at PF employs an APPLE-II type undulator while ALS beamline 5.3.2.2 is illuminated by bending magnet radiation. The desired x-ray photon energies from this broadband source are selected with a monochromator grating and focused to a fixed ~35 nm beam spot using a Fresnel zone plate[52]. Prior to the experiments, the microscope energy control was calibrated at beamline 19A using highly ordered pyrolytic graphite and at beamline 5.3.2.2 using the sharp C and O photoabsorptions of $CO_2$ gas introduced into the STXM chamber. The different calibration standards resulted in an energy shift of 0.2 eV between spectra acquired at the two facilities. During the experiments, the chambers for both beamlines were backfilled with He to ~30 kPa (0.3 bar) to provide temperature stability. Beamline 5.3.2.2 also includes a $N_2$ gas filter, designed to block higher order interference from oxygen emission. This filter was used for initial measurements, but was removed after it was determined that nitrogen abundance was too low to affect carbon measurements. In hindsight, however, this choice is not optimal for in situ experiments (although it is valid for IOM samples), where organic grains could be closely associated with O-bearing silicate minerals, and some evidence for such interference was observed in subsequent Ryugu datasets, as a ubiquitous broad photoabsorption in C-XANES spectra.

X-ray absorption images were acquired at spatial resolutions up to 25 nm/pixel with a 2–3 ms dwell time per pixel, controlled by motion of the piezo stage motors. However, hyperspectral data "stacks" were acquired at 50 nm /pixel resolutions to maximize signal-to-noise while minimizing total photon dose. Locations of discrete carbonaceous grains in these stacks are shown in Supplementary Fig. 1. In the near-edge structure region of the energy range, energy steps were set at 0.1 eV resolution for carbon stacks, and 0.2 eV for nitrogen and oxygen stacks. The optical density of each image in the data stack is calculated from background absorption ($I_0$) by $OD = -\log(I/I_0)$. XANES spectra of carbonaceous regions of interest are generated by summing pixels in those regions in the optical density datasets. PF STXM data were processed using the Python package Mantis[53] and custom Python scripts[8]. Individual XANES spectra were deconvolved by fitting to a series of twenty Gaussian peaks of fixed position and width (allowing for peak height to vary)[31], and a Python-based hierarchical, agglomerative clustering algorithm using the Ward variance minimization metric was applied to this parameterized, 20-dimensional dataset to identify clusters of similar spectral shapes (Fig. 1D and Supplementary Figs. 5, 6).

## Scanning transmission electron microscopy

High angle annular dark field (HAADF) and bright field STEM images in this study were obtained with at the Naval Research Laboratory with a Nion UltraSTEM 200-X operated at 60 kV, with a nominal probe size of 0.15 nm and probe currents of 50–100 pA. Samples were baked at 140 °C to maintain the ultra-high vacuum of the microscope.

## Secondary ion mass spectrometry

The FIB section extracted from Ryugu grain A0108-3 was returned to the FIB, where it was removed from the TEM grid and welded with Pt flat onto a clean Au foil. A thin Au coating was applied to avoid surface charging effects, and the sample was analyzed with the Cameca NanoSIMS 50 L at the Carnegie Institution of Washington. A focused $Cs^+$ beam was rastered across the section with simultaneous collection of negative ions and secondary electrons. A large beam (hundreds of pA) was first used to pre-sputter the section to remove the Au coating and any surface contamination until stable secondary ion currents were obtained. A ~0.4 pA beam, at <100 nm diameter, was then rastered (256 × 256 pixels over a 10 μm × 10 μm region, 100 image frames of 65 s each) over the sample with simultaneous collection of negative secondary ions of $^{16}O$, $^{12}C_2$, $^{12}C^{13}C$, $^{12}C^{14}N$, $^{12}C^{15}N$, $^{28}Si$, and $^{32}S$. The NanoSIMS ion images were analyzed with the L'image software (L. Nittler). After ion images were corrected for detector deadtime and aligned, isotopic ratio images were generated and C-rich regions of interest were defined through a combination of automatic image segmentation and manual selection, based on C grains identified by the previous STXM and STEM investigations. Synthetic $Si_3N_4$ grains, as well as IOM from a CR chondrite, were used to correct the data for instrumental fractionation.

# Data availability

STXM-XANES spectra reported in this paper were collected as part of the Hayabusa2 Initial Analysis Team activities, and have been deposited in the Data Archives and Transmission System (DARTS) at https://

data.darts.isas.jaxa.jp/pub/hayabusa2/paper/sample/DeGregorio_2024/. Raw STXM data acquired at the Advanced Light Source are protected and are not available due to U.S. Department of Energy data privacy laws. However, raw STXM data from this study are available upon request from the beamline scientist (Matthew Marcus; mamarcus@lbl.gov).

## Code availability

STXM datasets were processed using custom Python scripts developed and reported by Hikaru et al[8]. Spectrum shape analysis used the standard hierarchical clustering package from `scipy.cluster.heirarchy`, available in most Python implementations.

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

## Acknowledgements

Co-author A.L.D.K. passed away during this study, who provided essential contributions to STXM/XANES acquisition and analysis. This research used resources of the Advanced Light Source, a U.S. DOE Office of Science User Facility under contract no. DE-AC02-05CH11231. Synchrotron beamline scientist Matthew Marcus assisted in access, operation and analysis of STXM data collected at the Advanced Light Source. U.S. Naval Research Laboratory efforts in this study were supported by the NASA Solar System Exploration Research Virtual Institute node RIS2E (80HQTR20T0014) to B.T.D. and a NASA LARS award (80HQTR20T0050) to R.M.S. C.L. was funded by ISITE ULNE and the "Métropole Européenne de Lille" through the "TEM-Aster projet". Carnegie Institution efforts were supported by a NASA Hayabusa2 Participating Scientist award (NNX16AK72G) to L.R.N. Japanese efforts in this study were supported by KAKENHI from the Japan Society for the Promotion of Science (JSPS) (grant nos. JP20H05846, 19H01954, 18H04461, 20H04615 to H.Y. and grant no. JP17H06458 to Y.K., S.Y., Ya.T., and Yo.T. to support the STXM beamline 19A at Photon Factory), the Astrobiology Center Program of National Institutes of Natural Sciences (NINS) (Grant Number AB312007, AB022001 and AB032004 to H.Y.) and the JSPS Core-to-Core program "International Network of Planetary Sciences". L.R. was supported by the European Research Council through the consolidator grant HYDROMA (grant agreement no. 819587). We also acknowledge helpful discussions with Bernard Marty and Katherine Burgess regarding data analysis and interpretation.

## Author contributions

B.T.D., G.D.C., R.M.S., A.L.D.K., S.S., C.L., L.R.N., J.B., H.Y., Z.M. and Y.K. were involved with, data accumulation, data processing, writing and editing the manuscript text. These authors and T.O., M.H., S.Y., Ya.T., Yo.T., D.W., C.E., L.Be., L.Bo., E.Q., L.R., J.D., M.V.-P., S.M., M.K., J.M., A.D., A.D.-B., E.D., Yu.Ta., H.S., G.M., K.K., M.S., M.M. and Y.E. were involved with experimental design and sample preparation. M.Y., T.S., Sa.T., F.T., S.N., T.U., M.A., T.O., T.Y., M.N., A.N., A.M., K.Y. were involved with curation and handling of the asteroid grains at JAXA. H.Y., T.Na., T.No., R.O., H.N., K.S., Sh.T., S.W. and Yu.Ts. were the Executive Council of the Hayabusa2 Initial Analysis Team.

## Competing interests

The authors declare no competing interests.

## Additional information

Bradley De Gregorio[1]✉, George D. Cody[2], Rhonda M. Stroud[3], A. L. David Kilcoyne[4], Scott Sandford[5], Corentin Le Guillou[6], Larry R. Nittler[3], Jens Barosch[7], Hikaru Yabuta[8], Zita Martins[9], Yoko Kebukawa[10], Taiga Okumura[11], Minako Hashiguchi[12], Shohei Yamashita[13], Yasuo Takeichi[13], Yoshio Takahashi[11], Daisuke Wakabayashi[13], Cécile Engrand[14], Laure Bejach[14], Lydie Bonal[15], Eric Quirico[15], Laurent Remusat[16], Jean Duprat[16], Maximilien Verdier-Paoletti[16], Smail Mostefaoui[16], Mutsumi Komatsu[17], Jérémie Mathurin[18], Alexandre Dazzi[18], Ariane Deniset-Besseau[18], Emmanuel Dartois[19], Yusuke Tamenori[20], Hiroki Suga[21], Gilles Montagnac[22], Kanami Kamide[8], Miho Shigenaka[8], Megumi Matsumoto[23], Yuma Enokido[23], Makoto Yoshikawa[24], Takanao Saiki[24], Satoshi Tanaka[24], Fuyuto Terui[25], Satoru Nakazawa[24], Tomohiro Usui[24], Masanao Abe[24], Tatsuaki Okada[24], Toru Yada[24], Masahiro Nishimura[24], Aiko Nakato[24], Akiko Miyazaki[24], Kasumi Yogata[24], Hisayoshi Yurimoto[26], Tomoki Nakamura[23], Takaaki Noguchi[27], Ryuji Okazaki[28], Hiroshi Naraoka[28], Kanako Sakamoto[24], Shogo Tachibana[11], Sei-ichiro Watanabe[12] & Yuichi Tsuda[24]

[1]Materials Science and Technology Division, U.S. Naval Research Laboratory, Washington, DC, USA. [2]Earth and Planets Laboratory, Carnegie Institution for Science, Washington, DC, USA. [3]School of Earth and Space Exploration, Arizona State University, Tempe, AZ, USA. [4]Advanced Light Source, Lawrence Berkeley National Laboratory, Berkeley, CA, USA. [5]NASA Ames Research Laboratory, Moffett Field, Mountain View, CA, USA. [6]Unité Matériaux et Transformations, Université de Lille, Villeneuve d'Ascq, France. [7]School of Geosciences, University of Edinburgh, Edinburgh, United Kingdom. [8]Department of Earth and Planetary Systems Science, Hiroshima University, Hiroshima, Japan. [9]Centro de Química Estrutural, Institute of Molecular Sciences and Department of Chemical Engineering, Instituto Superior Técnico, Universidade de Lisboa, Lisboa, Portugal. [10]Department of Earth and Planetary Sciences, Tokyo Institute of Technology, Tokyo, Japan. [11]Department of Earth and Planetary Science, University of Tokyo, Tokyo, Japan. [12]Department of Earth and Environmental Sciences, Nagoya University, Nagoya, Japan. [13]Photon Factory, High Energy Accelerator Research Organization, Tsukuba, Japan. [14]Laboratoire de Physique des 2 Infinis Irène Joliot-Curie, Université Paris-Saclay, Centre National de la Recherche Scientifique, Orsay, France. [15]Institute de Planétologie et d'Astrophysique, Université Grenoble Alpes, Grenoble, France. [16]Institut de Mineralogie, Physique des Materiaux et Cosmochimie, Museum National d'Histoire Naturelle, Centre National de la Recherche Scientifique, Sorbonne Université, Paris, France. [17]General Education Department, Saitama Prefectural University, Saitama, Japan. [18]Institut Chimie Physique, Université Paris-Saclay, Centre National de la Recherche Scientifique, Orsay, France. [19]Institut des Sciences Moléculaires d'Orsay, Université Paris-Saclay, Centre National de la Recherche Scientifique, Orsay, France. [20]Department of Chemistry, Tokyo Metropolitan University, Tokyo, Japan. [21]Japan Synchrotron Radiation Research Institute, Hyogo, Japan. [22]École normale supérieure de Lyon, University Lyon 1, Lyon, France. [23]Department of Earth Science, Tohoku University, Sendai, Japan. [24]Institute of Space and Astronautical Science, Japan Aerospace Exploration Agency, Sagamihara, Japan. [25]Department of Mechanical Engineering, Kanagawa Institute of Technology, Atsugi, Japan. [26]Department of Earth and Planetary Sciences, Hokkaido University, Sapporo, Japan. [27]Department of Earth and Planetary Sciences, Kyoto University, Kyoto, Japan. [28]Department of Earth and Planetary Sciences, Kyushu University, Fukuoka, Japan. ✉e-mail: bradley.t.degregorio.civ@us.navy.mil

