## [Peer Review File · Nature Communications]

Variations of Organic Functional Chemistry in Carbonaceous Matter from the Asteroid 162173 RyuguREVIEWER COMMENTS

Reviewer #1 (Remarks to the Author):

This paper reports on an excellent dataset from Ryugu organic matter (OM) acquired by a combination of XANES-STEM-NanoSIMS methods on extracted “insoluble OM” as well as on minimally processed Ryugu particles and their OM grains. The complementary information on functional chemistry and isotopic composition is of particular importance for this new type of extraterrestrial sample brought back from an asteroid. The authors claim that they may have located a new type of organic material (“alkyl-aromatic”, AA) within Ryugu OM, which is possibly characterized by a different functional chemistry, i.e., a narrow aromatic “285 eV” peak and a spectral “bump” in the 287-289 eV range due to more abundant carboxyl and aliphatic carbon bonding environments. If this is really the case, then this work is worth being published in Nature Communications, especially if the combination with isotopic C-N data indicates a different origin or evolution of this material.

However, at the moment, I do not really see that this is the case. In Fig. 1, the four OM grains of the AA type look indeed very different to the “highly aromatic” (HA) grains, but I do not see a clear distinction between “IOM-like” (IL) and AA grains. The 285 band of the IL grains is also a bit narrower compared with the HA type, and in some of the IL spectra the carboxyl band is also not very sharp or distinct, but more like a “bump” (e.g., grains A2 or B2). This is also seen in their quantification ternary diagram (Fig. 2), where the AA and IL grains overlap. It is also known from meteorite OM investigations that the third IOM band (~288 eV, “carboxylic”) can be highly variable among different samples and even among the same meteorite in differently altered lithologies.

Furthermore, although the authors state in the abstract that isotopic compositions vary between the three functional group types, I only recognize this in Fig. 3 for the HA and IL/AA grains, which seem to show this difference. The isotope data is based on a very limited dataset from one lamella, so if it is possible to verify that different types of OM bonding environments have different isotopic compositions, this would be also highly interesting.

They also argue that the OM types differ at the O K-edge, because the HA/AA grains show a different shape at 531 eV than the IL grains or bulk IOM. However, this figure is hidden in the supplementary file (Fig. S2), and if this information is important to underline this distinction it would be good to show in the main manuscript, maybe even with some further explanations.

They also state that “diffuse OM” in Ryugu samples might be more ubiquitous relative to meteorite samples (L. 388) to underline their assertion that Ryugu OM has sampled more pristine organics. However, I do not think that this is true, because it has been shown by several authors (Le Guillou, Vinogradoff, Changela, ...) that this diffuse material is widespread in meteorites, but simply more difficult to study than the more globular OM grains, indicating a sampling bias.

There are already some functional group investigations on Ryugu OM by, e.g., Yabuta+, Ito+, and very recently by Stroud+ using very similar techniques, and in none of that work is there some indication for a new type of OM material with a different spectral shape. The only exception is the highly aliphatic material found by Ito+, but it is a bit surprising that the authors do not see strong indications of that band in their “AA” spectra, especially if they claim that the smaller aromatic units must be interconnected by longer aliphatic chains (L. 214).

To summarize, I would support publication of this paper in Nature Communication if the authors

could strengthen their hypothesis that this new AA-material is really unique to Ryugu based on, for example, additional functional group analyses or quantifications (O-K, ...) and/or more isotope data.

Reviewer #2 (Remarks to the Author):

The manuscript by De Gregorio et al. describes the results obtained from the analysis of various sample preparations of Ryugu material (IOM, thin sections of particle fragments prepared by ultramicrotomy, thin sections extracted by FIB) on carbonaceous matter. Different techniques were used for these characterizations: XANES spectroscopy, STEM, NanoSIMS. Three main results are discussed:

- 1) the detection of carbonaceous grains belonging to 3 spectral types (from XANES data). Two correspond to groups identified in previous studies while the third one displays a singular XANES spectral shape, that the authors tentatively interpret to be due to an increased amount of smaller polyaromatic domains, predominantly single or double benzene rings, decorated predominantly by carboxyl functional groups (referred to as “alkyl-aromatic” (AA) material).
- 2) A potential link between the functional chemistry of the 3 XANES spectral types identified and their isotopic composition, possibly reflecting distinct origins.
- 3) The presence of CO₃ functional groups in the diffuse carbonaceous material, associated with phyllosilicates, similar to CI/CM meteorites.

The manuscript is well written and the structure is clear. The results presented in the paper tend to extend previous analyses on Ryugu samples (e.g., Yabuta et al. 2023). The novelty of the results is, however, not always fully clear and would require some clarifications. The presentation of the results could also be improved in some cases, in particular for the XANES analyses. I detail my different comments below:

- The description of the three XANES spectral types in the first part would require some clarifications. Some of the spectral differences described in the text are not clearly visible in Figure 1. While I understand why the spectra are shown with an offset, this makes it very difficult to assess possible differences between the spectral types. I would suggest to have an additional figure with spectra of the different types overlapped to make a proper comparison. I would also strongly suggest to add vertical lines to clearly see the abscissas of the different peaks. I would also strongly recommend to add arrows that indicate the features describes in the text.

- To continue on my previous comment, looking at Figure 1E, I'm not convinced that the 3rd kind of spectra (AA), presented by the authors as novel, is really different from the “IOM-like”. I thus strongly encourage the authors to clarify/improve the Figure.

- Figure 2: can you clarify why green and purple color fields overlap?

- The authors claim that correlated STXM, STEM, and NanoSIMS data suggest that there is a link between the functional chemistry of the 3 XANES spectral types identified and their isotopic

composition. As mentioned by the authors, the dataset is really limited, and the results shown in Figure 3 and highlighted in the text (“Both the highest ^{15}N and ^{13}C enrichments are observed in grains with an IL functional group chemistry, while the two measured grains with HA functional chemistry contain depletions in both ^{15}N and ^{13}C ”) do not appear sufficient for such a claim. The large variations of ^{15}N and ^{13}C have already been observed in previous studies (e.g. Figure 6 of Yabuta et al. 2023) and discussed. It is not clear what is new here and what how does that lead to the conclusion that there is a link between the functional chemistry and isotopic composition.

- Concerning the identification of CO_3 rich material, it displays a 290.4eV peak assigned to CO_3 functional groups, but lacks a $\text{C}=\text{O}$ peak. In parallel, the authors rule out carbonate minerals as there is a lack of a characteristic, strong, extended fine structure typically observed in A0108-3 (Figure 4). Can you clarify if the latter (due to carbonate minerals) could be also “hidden”, similar to the $\text{C}=\text{O}$ peak or because of a different process? In particular, such 290.4eV peak has also been observed by Yabuta et al. 2023. Similarly, no fine structure at higher energies (294 to 304 eV) could be observed, and Yabuta et al. concluded that the carbonates were not in a crystalline structure but more likely present as molecular carbonate. They also suggested that the Mg signature observed on clay-bound organics could indicate that Mg “be associated with molecular CO_3 , MgCO_3 , or a combination of these”.

- “and visual TEM confirmation of organic matter in between phyllosilicate sheets (Figure 4D) indicates the presence of an abundant clay-associated polycarbonate material in Ryugu.” => I think Figure 4C should be referenced instead of Figure 4D.

- Figure 4C. I would suggest to add some information on the image and the label to clarify what we observe on the image and why it indicates the presence of an abundant clay-associated polycarbonate material.

Response to Reviewers – NCOMMS-24-03273A

Response to Reviewer #1:

This paper reports on an excellent dataset from Ryugu organic matter (OM) acquired by a combination of XANES-STEM-NanoSIMS methods on extracted “insoluble OM” as well as on minimally processed Ryugu particles and their OM grains. The complementary information on functional chemistry and isotopic composition is of particular importance for this new type of extraterrestrial sample brought back from an asteroid. The authors claim that they may have located a new type of organic material (“alkyl-aromatic”, AA) within Ryugu OM, which is possibly characterized by a different functional chemistry, i.e., a narrow aromatic “285 eV” peak and a spectral “bump” in the 287-289 eV range due to more abundant carboxyl and aliphatic carbon bonding environments. If this is really the case, then this work is worth being published in Nature Communications, especially if the combination with isotopic C-N data indicates a different origin or evolution of this material.

However, at the moment, I do not really see that this is the case. In Fig. 1, the four OM grains of the AA type look indeed very different to the “highly aromatic” (HA) grains, but I do not see a clear distinction between “IOM-like” (IL) and AA grains. The 285 band of the IL grains is also a bit narrower compared with the HA type, and in some of the IL spectra the carboxyl band is also not very sharp or distinct, but more like a “bump” (e.g., grains A2 or B2). This is also seen in their quantification ternary diagram (Fig. 2), where the AA and IL grains overlap. It is also known from meteorite OM investigations that the third IOM band (~288 eV, “carboxylic”) can be highly variable among different samples and even among the same meteorite in differently altered lithologies.

The reviewer is correct in that the differences between AA and IL spectra are subtle. Except in the most obvious cases, most readers would not spot the differences. However, our hierarchical clustering algorithm consistently identified clusters with the AA spectral shape that are distinct from the IL spectral shapes. Both macromolecular structures are similar in that they are polyaromatic domains interconnected by aliphatic side chains, and modified by various functional groups. But the AA spectra are unique due to their intense and sharp 285 eV absorption and increase in aliphatic content.

We have softened our language regarding the distinction of these two spectral shapes throughout the paper. We have also removed all mentions of the AA shape being novel or new, as it has already been reported in Yabuta et al. (2023). The reviewer mentions the ternary diagram in Figure 2 where the color fields overlap for the IL and AA spectra, and we have added the following text to address this overlap:

“Plotting the spectral contributions from the major absorption features in these spectra show a clear distinction between HA and IL spectral shapes, as well as the related cluster of AA grains (Figure 2E). The overlap between the AA and IL fields in this plot illuminate how similar these two spectral shapes are to the human observer, and other non-traditional representations of C-XANES spectra may be necessary to distinguish them (e.g., Suppl. Figure S7).”

Figure S7 shows a comparison of peak height and peak width of the aromatic absorption at 285 eV, in which AA and IL spectra have separate fields.

We have also added more explicit mentions of our hierarchical clustering methods for identifying spectral shape, to make clear that we are not relying on human identification of spectral shapes:

“We used spectral decomposition with a series of Gaussian peaks and a hierarchical clustering algorithm to perform a shape analysis and identify similarities within the spectral dataset (Methods; Suppl. Figure S2).”

“Hierarchical clustering identified a group of four (~20%) Ryugu IOM grains distinct from the typical IL and HA spectral shapes.”

Finally, concerning the variability of the carboxyl band, we refer to the paper by Urquhart and Ade (2002), who provided a thorough investigation of the photoabsorption of the C=O functional group in various molecules. They found that there was a gap in the range between 286.7 and 288 eV, in both their experimental and ab initio results. While we cannot rule out a more complicated carbonyl-bearing functional group absorption in this energy region, we find it more likely that our interpretation of aliphatic photoabsorption in this energy region to be more likely.

Furthermore, although the authors state in the abstract that isotopic compositions vary between the three functional group types, I only recognize this in Fig. 3 for the HA and IL/AA grains, which seem to show this difference. The isotope data is based on a very limited dataset from one lamella, so if it is possible to verify that different types of OM bonding environments have different isotopic compositions, this would be also highly interesting.

It is unfortunately that this is the only data available to us from the Hayabusa2 Initial Analysis period. We intend to continue collecting correlated NanoSIMS and C-XANES data on Ryugu samples, but those datasets will not be completed until Fall 2024. We have used careful language in our manuscript that the current data merely suggests the likelihood of a connection between functional chemistry and isotopic composition of carbonaceous grains in Ryugu, and have added a “may” in one instance to further soften our implication. We hope that the editors accept this response so as to not delay publication further.

They also argue that the OM types differ at the O K-edge, because the HA/AA grains show a different shape at 531 eV than the IL grains or bulk IOM. However, this figure is hidden in the supplementary file (Fig. S2), and if this information is important to underline this distinction it would be good to show in the main manuscript, maybe even with some further explanations.

After careful reanalysis of our O-XANES data, we have concluded that the few spectra we have do not show enough of a distinction between the IL and AA spectral shapes. Most of our O-XANES data are acquired from microtome samples, for which contributions from the surrounding silicate minerals play an outsized role. Therefore we have removed the text describing differences in O-XANES spectra and replaced it with:

“O-XANES spectra of Ryugu IOM, IL grains, and this third spectral type show a similar intensity ratio between the 531 eV π^* absorption from carbonyl C=O functional groups and the σ^* absorption at 540 eV from C-O single bonds³⁴ (Suppl. Figure S4), suggesting relatively similar abundances of O composition. In contrast, HA grains show a smaller $\pi^*:\sigma^*$ peak ratio due to a smaller abundance of O-bearing functional groups.”

Because the O-XANES data do not play a significant role in the discussion anymore, we have opted to keep this figure in the Supplementary Materials.

They also state that “diffuse OM” in Ryugu samples might be more ubiquitous relative to meteorite samples (L. 388) to underline their assertion that Ryugu OM has sampled more pristine organics. However, I do not think that this is true, because it has been shown by several authors (Le Guillou, Vinogradoff, Changela, ...) that this diffuse material is widespread in meteorites, but simply more difficult to study than the more globular OM grains, indicating a sampling bias.

We disagree with this comment. Our data, and shown in Figure 4B, reveal that diffuse organic matter gives strong photoabsorption across all FIB and microtome sections, indicating a high abundance. We simply do not ever observe such high concentrations of diffuse organics in meteorites samples, and this is seen in the papers the reviewer mentions, where only small patches of diffuse organics are identified. To support our conclusion, we have added the following text:

“XANES spectral mapping of this 290.4 eV peak in FIB and microtome sections reveals relatively large regions of diffuse organic matter (Figure 4B), which is not similarly observed in CI or CM chondrites.”

There are already some functional group investigations on Ryugu OM by, e.g., Yabuta+, Ito+, and very recently by Stroud+ using very similar techniques, and in none of that work is there some indication for a new type of OM material with a different spectral shape. The only exception is the highly aliphatic material found by Ito+, but it is a bit surprising that the authors do not see strong indications of that band in their “AA” spectra, especially if they claim that the smaller aromatic units must be interconnected by longer aliphatic chains (L. 214).

Our results are most similar to that of Yabuta et al. (2023), and that study describes two new types of organic matter based on XANES data. The first is called “aromatic”, which matches what we call “alkyl-aromatic” (or AA type) in our study. The second is called “molecular carbonate” by Yabuta et al., which we also identify in our dataset and also call “molecular carbonate”. We have added a sentence in the text to be more clear that we are not purporting to have discovered yet another type of organic matter in Ryugu:

“Our finding that the XANES spectra from discrete grains cluster into three main spectral shapes matches the preliminary results from the Hayabusa2 Initial Analysis, where in that study the AA spectral shape is identified as “aromatic”¹⁴.

The data in Ito et al. (2022) is quite interesting. We were looking to find similar spectra in our samples, but we did not. I have analyzed that STXM data shown in Ito et al., and can reproduce their results. I have seen other presentations at conferences that suggest even one more additional aliphatic-rich spectral type as well. As researchers investigate more Ryugu samples, I believe we will see more published examples of exotic XANES spectral shapes. But that is outside the scope of our present study.

To summarize, I would support publication of this paper in Nature Communication if the authors could strengthen their hypothesis that this new AA-material is really unique to Ryugu based on, for example, additional functional group analyses or quantifications (O-K, ...) and/or more isotope data.

We would like to clarify that although we find that AA organic matter appears to be unique to the Ryugu asteroid, we are not suggesting that it cannot exist in the related CI chondrites as well. Future work on CI chondrites may reveal AA grains that would not have been recognized due to their similarities to IL spectral shapes. We have added to the following text:

“Application of automated spectral shape analysis to future studies of organic matter in CI chondrites may also discover examples of AA grains in these Ryugu analog samples.”

We believe that the changes we have made to the text and figures are sufficient to merit publication in *Nature Communications*.

Response to Reviewer #2:

The manuscript by De Gregorio et al. describes the results obtained from the analysis of various sample preparations of Ryugu material (IOM, thin sections of particle fragments prepared by ultramicrotomy, thin sections extracted by FIB) on carbonaceous matter. Different techniques were used for these characterizations: XANES spectroscopy, STEM, NanoSIMS. Three main results are discussed:

1) the detection of carbonaceous grains belonging to 3 spectral types (from XANES data). Two correspond to groups identified in previous studies while the third one displays a singular XANES spectral shape, that the authors tentatively interpret to be due to an increased amount of smaller polyaromatic domains, predominantly single or double benzene rings, decorated predominantly by carboxyl functional groups (referred to as “alkyl-aromatic” (AA) material).

2) A potential link between the functional chemistry of the 3 XANES spectral types identified and their isotopic composition, possibly reflecting distinct origins.

3) The presence of CO₃ functional groups in the diffuse carbonaceous material, associated with phyllosilicates, similar to CI/CM meteorites.

The manuscript is well written and the structure is clear. The results presented in the paper tend to extend previous analyses on Ryugu samples (e.g., Yabuta et al. 2023). The novelty of the results is, however, not always fully clear and would require some clarifications. The presentation of the results could also be improved in some cases, in particular for the XANES analyses. I detail my different comments below:

We thank the reviewer for their comments. Our study is indeed an extension of the work by Yabuta et al. (2023), for which many of the authors of the present paper also contributed. The present study includes new data, which supports the original conclusions of Yabuta et al. (2023).

- The description of the three XANES spectral types in the first part would require some clarifications. Some of the spectral differences described in the text are not clearly visible in Figure 1. While I understand why the spectra are shown with an offset, this makes it very difficult to assess possible differences between the spectral types. I would suggest to have an additional figure with spectra of the different types overplotted to make a proper comparison. I would also strongly suggest to add vertical

lines to clearly see the abscissas of the different peaks. I would also strongly recommend to add arrows that indicate the features describes in the text.

We have added lines to Figure 1E to denote the locations of the photoabsorptions for aromatic carbon (C=C), ketone (C=O), and carboxyl (COOH) functional groups. Text annotations of these lines identify them without the need to add arrows. We have also added a new figure to the supplementary information showing overlapping spectra from each of the three main spectral types, along with a plot of these spectra differences to Ryugu IOM, showing where each spectral types have more or less absorption relative to IOM.

Text additions related to this comment about the similarities between IL and AA spectral shapes can be found in our responses to the first reviewer.

- To continue on my previous comment, looking at Figure 1E, I'm not convinced that the 3rd kind of spectra (AA), presented by the authors as novel, is really different from the "IOM-like". I thus strongly encourage the authors to clarify/improve the Figure.

This comment is an extension of the previous comment. To address this, we have added a new figure to the supplementary information showing spectral differences between the different spectral shapes. In this plot, there is excess intensity (x-ray absorption) for the alkyl-aromatic spectral shape between 287-288 eV.

- Figure 2: can you clarify why green and purple color fields overlap?

This comment was also pointed out by the first reviewer and has been addressed by our previous response to said reviewer.

- The authors claim that correlated STXM, STEM, and NanoSIMS data suggest that there is a link between the functional chemistry of the 3 XANES spectral types identified and their isotopic composition. As mentioned by the authors, the dataset is really limited, and the results shown in Figure 3 and highlighted in the text ("Both the highest 15N and 13C enrichments are observed in grains with an IL functional group chemistry, while the two measured grains with HA functional chemistry contain depletions in both 15N and 13C") do not appear sufficient for such a claim. The large variations of 15N and 13C have already been observed in previous studies (e.g. Figure 6 of Yabuta et al. 2023) and discussed. It is not clear what is new here and what how does that lead to the conclusion that there is a link between the functional chemistry and isotopic composition.

We believe the language we use in this paper establishes that we are merely reporting a possible connection between functional chemistry and isotopic composition. We have added text to this paragraph to clarify that we are tying the C-XANES data (new) to existing NanoSIMS data, emphasizing what is new in our study:

"Previous studies of Ryugu samples have indicated a large spread of ^{15}N , ^{13}C , and D values, but not correlated with functional chemistry of the organic matter. Here we report C-XANES data of 15 relatively large carbonaceous grains for which N and C isotopic compositions were previously measured (Figure 3)¹⁴. Most of the measured carbonaceous grains in this section (small gray dots in Figure 3E), identified by

concentrations of carbonaceous matter in NanoSIMS image frames, have C and N isotopic compositions indistinguishable from the “bulk” isotopic composition of Ryugu organic matter^{14,36}, within errors. However, most of the large carbonaceous grains deviate by $>2\sigma$ in at least one of these isotopic systems (Suppl. Table S1).”

Collection of additional data would take many additional months, and therefore we hope that the editors accept our changes so as to not delay publication.

- Concerning the identification of CO₃ rich material, it displays a 290.4eV peak assigned to CO₃ functional groups, but lacks a C=O peak. In parallel, the authors rule out carbonate minerals as there is a lack of a characteristic, strong, extended fine structure typically observed in A0108-3 (Figure 4). Can you clarify if the latter (due to carbonate minerals) could be also “hidden”, similar to the C=O peak or because of a different process? In particular, such 290.4eV peak has also been observed by Yabuta et al. 2023. Similarly, no fine structure at higher energies (294 to 304 eV) could be observed, and Yabuta et al. concluded that the carbonates were not in a crystalline structure but more likely present as molecular carbonate. They also suggested that the Mg signature observed on clay-bound organics could indicate that Mg “be associated with molecular CO₃, MgCO₃, or a combination of these”.

The reviewer is correctly pointing out the similarity between the CO₃-rich organics we describe in our paper and that described by Yabuta et al. (2023) and Stroud et al. (2024). They are the same organic matter. We have added statements to the manuscript that make clearer that we are observing the same material:

“This abundant molecular carbonate was also previously reported in Ryugu matrix^{14,41}.”

The reviewer makes an interesting counterpoint that mineral carbonate EXAFS peaks could be hidden somehow. I can speak from experience that even nanoscale carbonates show clear EXAFS peaks in C-XANES spectra. We have edited the text to include this point:

“This extended fine structure is present in a calcite grain is observed in the section and within organic matter in the immediate surrounding mineral matrix, presumably taking the form of nanoscale calcite grains.”

We have also added extra references to the Yabuta et al. (2023) and Stroud et al. (2024) studies.

- “and visual TEM confirmation of organic matter in between phyllosilicate sheets (Figure 4D) indicates the presence of an abundant clay-associated polycarbonate material in Ryugu.” => I think Figure 4C should be referenced instead of Figure 4D.

This has been corrected.

- Figure 4C. I would suggest to add some information on the image and the label to clarify what we observe on the image and why it indicates the presence of an abundant clay-associated polycarbonate material.

We have added arrows to Figure 4C denoting the locations of interstitial organic matter associated with the phyllosilicate sheets. Accordingly, we have updated the caption as well:

“(C) Bright-field STEM image of intermixed phyllosilicate sheets and brighter, interstitial organic matter from the region shown in (B). Arrows denote locations of accumulations of CO³-rich organic matter.”

REVIEWERS' COMMENTS

Reviewer #1 (Remarks to the Author):

The authors have done a great job in improving the manuscript to make their assignments clearer to the reader, specifically concerning the new "AA" type of organic matter. It is also good that they have softened their language accordingly. I still believe that the diffuse type of OM is more widespread in meteoritic samples than the authors claim due to the aforementioned sampling bias, but this discussion is beyond the scope of the paper. I am generally fine with their statement "..., which is not similarly observed in CI or CM chondrites.", but maybe they can add something like "...but more work on this type of diffuse OM is needed in meteorite samples". I therefore support publication in Nature Communications.